# A Comparison of Three Perfusion Algorithms in Patients at Risk of Delayed Cerebral Ischemia After Subarachnoid Hemorrhage

**DOI:** 10.3390/diagnostics15172236

**Published:** 2025-09-03

**Authors:** Lea Katharina Falter, Dirk Halama, Cordula Scherlach, Felix Arlt, Kristin Starke, Karl-Titus Hoffmann, Cindy Richter

**Affiliations:** 1Department of Neuroradiology, Leipzig University Hospital, Liebigstr. 20, 04103 Leipzig, Germany; lea.falter@gmail.com (L.K.F.); cordula.scherlach@medizin.uni-leipzig.de (C.S.); karl-titus.hoffmann@medizin.uni-leipzig.de (K.-T.H.); 2Department of Oral and Maxillofacial Surgery, Leipzig University Hospital, Liebigstr. 20, 04103 Leipzig, Germany; dirk.halama@medizin.uni-leipzig.de; 3Department of Neurosurgery, Leipzig University Hospital, Liebigstr. 20, 04103 Leipzig, Germany; felix.arlt@medizin.uni-leipzig.de; 4Department of Neurology, Leipzig University Hospital, Liebigstr. 20, 04103 Leipzig, Germany; kristin.starke@medizin.uni-leipzig.de

**Keywords:** delayed cerebral ischemia, CT perfusion, subarachnoid hemorrhage

## Abstract

**Background/Objectives**: Delayed cerebral ischemia (DCI) after an aneurysmal subarachnoid hemorrhage (aSAH) often presents with bilateral vasospasm and cortical spreading depolarizations. Computer tomography perfusion (CTP) is the prevailing screening method for detecting early changes in the cerebral blood flow. Commonly used CTP thresholds include an rCBF < 30% for the core volume and a Tmax > 6 s for hypoperfused tissue detection in acute ischemic stroke. These stroke algorithm computing thresholds compared to the contralateral hemisphere may or may not apply to detect tissue at risk of DCI. We aimed to quantify the volumetric agreement of three different stroke algorithms compared to the final infarct volumes as the standard. **Methods**: Furthermore, 123 CTP datasets of 75 patients with aSAH suspicious of DCI were processed using Intellispace Portal (ISP), Cercare Threshold, and Cercare Artificial Intelligence (AI) to calculate the tissue-at-risk (hypoperfused) and non-viable tissue (core) volumes. CT infarct volumes in plain CTs were segmented in the follow-up study by using a 3D slicer. **Results:** The calculated core volumes corresponded best to the final infarct volumes if DCI-related treatment was performed subsequently. Additional postprocessing improved the calculation of core volumes but overestimated the tissue at risk of hypoperfusion in DCI. Whereas the accuracy of tissue-at-risk prediction accelerated without treatment, underlining the importance of intra-arterial spasmolysis and induced hypertension in the prevention of DCI. **Conclusions:** Cercare AI and ISP revealed a sensitivity of 100% each, with a serious low specificity of <5% that was independent of treatment. Overall, the Cercare Threshold, applying the commonly used stroke thresholds, performed the best in predicting tissue at risk of hypoperfusion in DCI.

## 1. Introduction

Aneurysmal subarachnoid hemorrhage (aSAH) affects six to nine people per 100,000 per year and leaves more than 30% with lasting disabilities, often related to cognitive dysfunction [1,2]. The management of these patients is based on a limited number of randomized clinical trials [3,4,5,6,7,8]. One major cause of morbidity and mortality following an aSAH is delayed cerebral ischemia (DCI), with an overall prevalence of 0.29. It develops on days four to fourteen in the post-SAH phase and may result in cerebral infarction and a poor outcome. The pathogenesis of DCI is multifactorial, comprising cortical spreading depolarization, inflammatory reactions, microthromboembolism, autoregulatory failure, and micro- or macrovascular spasms [9].

aSAH patients with sudden-onset focal neurological deficits or unexplained altered consciousness receive regularly multimodal computed tomography (CT) scans, including a cranial CT, CT perfusion (CTP), and CT angiography (CTA). This non-invasive method is a commonly accepted screening method for DCI in unconscious patients. CTA has been reported to have 64% sensitivity and 96% specificity in assessing the location and severity of cerebral vasospasm after aSAH [10]. Postprocessing software generates perfusion maps immediately after data acquisition. Unlike transcranial Doppler, CTP can assess the blood flow in both the macro- and microvasculature. Cerebral vasospasm has classically been considered to be the only cause of DCI, but now, multiple concurrent and synergistic mechanisms have been suggested as the underlying pathologies [11]. Classical CTP algorithms, therefore, focus exclusively on preventing and treating reductions in blood flow supply. Impairment of the microvasculature and the collateral status have not been taken into account so far. Pathological brain activities characterized by hypersynchronous electrical disturbances, such as cortical spreading depolarization and seizures, can increase the cortical blood flow through neurovascular coupling to meet the rising energy demand [12]. Unlike acute vessel occlusions, cerebral vasospasm spreads from distal to proximal with an aggravation over days, compromising the chance to establish a collateral flow [13]. The concept of this analysis was the validation of the existing CTP algorithms for predicting DCI.

## 2. Materials and Methods

### 2.1. Population/Study Design and Recruitment

Approval from the local ethics committee (reference number: 412/19-ek) was obtained prior to conducting this study, and it was performed in accordance with the ethical standards of the Declaration of Helsinki.

We retrospectively identified 109 patients with acute SAH and CT perfusion datasets who presented to our hospital between January 2016 and March 2024 (Appendix A). The inclusion criteria were as follows: (1) an SAH verified by CT, MRI, or lumbar puncture; (2) digital subtraction angiography (DSA) with proof of a cerebral aneurysm; and (3) a CT perfusion dataset 0 to 21 days after a SAH. Thirty-four of these patients were excluded due to the following criteria: Traumatic SAH (*n* = 9), insufficient scan quality (*n* = 7), amyloid angiopathy (*n* = 1), cerebrospinal fluid infection (*n* = 2), arteriovenous malformation (*n* = 2), arterial dissection (*n* = 2), underlying malignancy (*n* = 1), and missing control CT (*n* = 10). The remaining 75 patients received one or more CT perfusion studies to detect DCI-related changes in cerebral perfusion.

### 2.2. Standard SAH Treatment Protocol

All patients were monitored in our neurointensive care unit. Nimodipine was given from the day of admission orally (6 × 60 mg/day) or via gastric tube. The mean arterial blood pressure was always sustained above 80 mmHg to avoid reduced cerebral perfusion. A local standard operating procedure for SAH was used (not included).

### 2.3. DCI

DCI was defined as a cerebral infarction that was identified on CT or MRI after excluding procedure-related infarctions. These infarctions had to occur within a time frame of at least 48 h but no later than six weeks after a SAH. Additionally, in accordance with Vergouwen et al. [14], our definition of DCI included the occurrence of focal neurological impairments, such as hemiparesis, hemihypesthesia, aphasia, apraxia, hemianopsia, or neglect, or a decrease of at least two points on the Glasgow coma scale. These deficits are not immediately present after aneurysm occlusion and cannot be attributed to other causes.

### 2.4. DCI Monitoring

DCI monitoring included daily transcranial Doppler (TCD) investigations of cerebral arteries. Cerebral vasospasm was suspected based on an altered level of consciousness or new neurological impairment (*n* = 77) or increased mean flow velocity in the TCD that exceeded 160 cm/s in the middle cerebral artery (MCA) or a doubling within 24 h (*n* = 15). In total, 123 multimodal CT examinations were performed to detect perfusion deficits or new territorial infarction. CT perfusion imaging included both initial screening scans (*n* = 13) and follow-up assessments (*n* = 18) after therapeutic interventions.

### 2.5. CT Perfusion

CT perfusion imaging was performed using a multislice helical 128-detector row CT scanner (Philips Ingenuity) with 4 cm axial coverage. Scanning consisted of an initial non-contrast head CT scan followed by CT perfusion and CTA acquisitions. For CT perfusion, 40 mL of contrast medium (Accupaque 350) followed by a 30 mL saline bolus at the same flow rate (4 mL/s) was applied. We used a scanning volume of 8 cm, which was obtained by double measurements (2 × 4 cm) and consisted of 16 sections at a 5 mm thickness. The acquisition parameters were 80 kVp and 100 mA, with a dose-length product of 474 mGy*cm. CT scanning was initiated 4 **s** after the start of the contrast bolus injection. Rapid imaging of the brain was performed repeatedly as the contrast bolus flowed through the brain, and the relative increase, peak, and then decrease in density, measured in Hounsfield units (HU), allowed a time–attenuation curve (TAC) to be derived for each voxel.

### 2.6. Postprocessing

Postprocessing of the acquired images was performed on two workstations, the Intellispace Portal (ISP, Philips Medical Systems, Best, The Netherlands) and Cercare (Cercare Medical, Copenhagen, Denmark). Motion correction was applied before postprocessing in each software, as even slight motion can degrade CT perfusion raw data. In order to create brain perfusion maps, time–density curves (TDC) are measured in an artery, vein, and at each voxel in the brain tissue, allowing for the visualization of contrast-labeled blood uptake. Tracer kinetic models are then used to estimate hemodynamic parameters from the 4-dimensional dataset.

The commonly used parameters are as follows:Cerebral blood volume (CBV, mL/100 g): the total volume of flowing blood per unit of brain mass.Cerebral blood flow (CBF, mL/100 g/min): the flow rate of blood through a unit of brain mass. Measurement of CBF is usually not quantitative but rather obtained by normalization to an unaffected region of the brain. Accordingly, CBF is expressed as a percentage compared to the reference ROI and is referred to as relative CBF (rCBF).Mean transit time (MTT, sec): the mean time for blood to perfuse a region of tissue. MTT is related to CBF and CBV by the central volume principle: MTT = CBV/CBF.Time to peak (TTP, sec): the time to the maximum point of the time–signal curve. It represents the time at which the maximum change in tracer concentration occurs after the passage of the bolus.Time to maximum (Tmax, sec): similar to TTP, Tmax reflects the time from the start of the scan until the maximum peak of contrast material in each voxel.

Each of these hemodynamic parameters is typically displayed as a color map of the brain, with a sequential scale representing the quantitative data values in each voxel. Automated postprocessing software further segments the brain parenchyma into normal tissue, tissue at risk (hypoperfused tissue), and non-viable tissue (core volume) based on quantitative or comparative parameter thresholds. We compared the following three CTP algorithms: Brain Perfusion of Intellispace Portal (ISP, Philips Medical Systems, Best, The Netherlands), Cercare Threshold (Cercare Medical, Copenhagen, Denmark), and Cercare Artificial Intelligence (AI, Cercare Medical, Copenhagen, Denmark).

### 2.7. Intellispace Portal

For postprocessing with ISP Brain Perfusion, the ROIs are placed over the terminal segment of the internal carotid artery for arterial input function (AIF) and the superior sagittal transverse sinus for venous output function (VOF) detection. This software uses a deconvolution method. Deconvolution provides the CBF, MTT, and Tmax. Other perfusion parameters can be estimated directly from the parenchymal TAC without deconvolution. ISP relies on CBV thresholds for the estimation of the ischemic core (rMTT > 150% and CBV > 2 mL/100 g) and hypoperfused volume (MTT > 150% and CBV < 2 mL/100 g). Relative MTT (rMTT) is the ratio between the MTT on both hemispheres.

### 2.8. Cercare

In the Cercare Medical software package, the AIF and VOF are automatically selected. Two algorithms are used to quantify the hypoperfused and core volume of ischemic strokes: Threshold and Artificial Intelligence (AI). Both algorithms use the same thresholds, including rCBF < 30% for the core volume and a Tmax of >6 s for hypoperfused tissue.

Threshold: In the threshold method, the core lesion is found by computing the mean CBF value in the entire contralateral hemisphere to the Tmax threshold lesion (Tmax > 6 s) and marking all voxels inside the original Tmax threshold lesion with a CBF value lower than 30% of the computed mean. The pathological side of the brain is identified as the Tmax threshold hypoperfused lesion. The mean rCBF value of the contralateral (healthy) side is calculated. It serves as a baseline.

Artificial intelligence: In the artificial intelligence (AI) method, the core lesion is found by mirroring the hypoperfusion lesion (Tmax > 6 s) to the contralateral hemisphere. Inside that “Mirror ROI”, the median CBF value is calculated within this volume. The median value is then used to mark all voxels inside the original hypoperfusion lesion with a CBF value < 30% of the computed median.

The generated Cercare maps include the following parameters:Oxygen extraction fraction (OEF, unitless): This represents the maximum oxygen extraction within a given voxel of brain tissue. This value depends on the distribution of transit time, which impacts oxygen extraction efficiency in the capillaries. In hypoperfusion lesions, OEF is typically higher as the tissue compensates for reduced blood flow by extracting more oxygen.Capillary transit time heterogeneity (CTH, sec): This represents the standard deviation of the TTD. Lower CTH values indicate homogenous transit times, whereas higher CTH values signify heterogenous transit times.

### 2.9. Standard

The infarct volume at the follow-up CT was used as the standard measure. The plain CT scan at the time-point of CTP (baseline) was compared to previous CT scans to exclude prior posthemorrhagic or ischemic strokes from measurements. Then, ischemic brain lesions in the baseline and follow-up CTs (following available CT scan) were delineated using a semi-automated technique (3D slicer v5.6.2, https://www.slicer.org/, accessed on 4 July 2024) by a neuroradiologist who was blinded to the clinical data and perfusion maps. A region-of-interest thresholding with manual adjustments was used to segment hypodense lesions in high-resolution CT scans with a slice thickness of 1.5 mm or less.

### 2.10. Statistical Analysis

Statistical analyses were performed with R version 4.5.1 (The R Foundation for Statistical Computing, Indianapolis, IN, USA). The determined hypoperfused and core volumes of the different algorithms, as well as the infarct volumes, showed a variance heterogeneity of means (*p* < 0.001). Therefore, Welch’s ANOVA and post hoc analysis (Games–Howell) were applied. A 2-tailed value of *p* < 0.05 indicated statistical significance.

## 3. Results

### 3.1. Study Group

Between January 2016 and March 2024, 385 patients with a SAH were admitted to our hospital. Among them, 109 patients underwent CT perfusion imaging. After applying the exclusion criteria, a final cohort of 75 patients undergoing a total of 123 CT perfusion scans was included in the analysis. The mean age of the study cohort was 56 years (range: 32–90 years). Half of the patient population presented with anterior communicating artery aneurysms (30.6%) and middle cerebral artery aneurysms (26.7%). Immediate aneurysm treatment was performed endovascularly (69.4%), surgically (25.3%), or combined (5.3%, Table 1).

### 3.2. CT Perfusion/DCI Treatment and Outcome

The median time from SAH onset to the first CTP scan was 5 days (range: 0–18 days). At the time of CTP imaging, 22.8% of scans were conducted in patients with a history of craniotomy, 38.2% demonstrated intracerebral hemorrhage, and 31.7% of assessed patients exhibited a Glasgow Coma Scale (GCS) below 7.

In our institutional clinical practice, perfusion deficits indicative of DCI were identified in 42 patients (56.0%). Management strategies included induced hypertension in 13 patients (17.3%) and intra-arterial spasmolysis in 29 patients (38.7%). The median number of intra-arterial spasmolysis procedures per treated patient was 2 (range: 1–10). Nine patients (31.0%) received Milrinone, seven patients (24.1%) were administered Nimodipine, and thirteen patients (44.8%) were treated with multiple agents. In 44.0% of all patients, no DCI-specific treatment was required. Among the patients who were treated with induced hypertension, 33.3% developed DCI–related irreversible infarction compared with 34.4% in the spasmolysis group and 35.3% in those who did not qualify for DCI-specific treatment. Overall, thirty-two patients (42.7%) developed irreversible infarction associated with delayed cerebral ischemia (DCI). Twenty-six patients (34.7%) suffered hemorrhagic infarctions.

Clinical outcomes were evaluated using the modified Rankin Scale (mRS) six months post-treatment. A favorable outcome (mRS 0–3) was observed in 39 patients (52.0%), while 36 patients (48.0%) had an unfavorable outcome (mRS 4–6). Detailed patient characteristics, treatment approaches, and outcomes are summarized in Table 1.

### 3.3. Four Categories of Prediction

To calculate sensitivity, specificity, positive predictive value (PPV), and negative predictive value, four categories of prediction were chosen (Table 2, Appendix A):No perfusion deficit/no infarct development: no abnormality;No perfusion deficit/infarct development: underestimated progressive infarct;Perfusion deficit/no infarct development: reversible perfusion deficit;Perfusion deficit/infarct development: progressive infarct.

Hypoperfused and core volumes were grouped according to volumes of 0 mL or greater. Final infarct volumes of the follow-up CTs were equally grouped into volumes of 0 mL or greater.

### 3.4. Overall Tissue Prediction

The overall distribution of the mean values of the core and hypoperfused volumes of each algorithm in correlation to the final infarct volumes is displayed in Figure 1.

### 3.5. Non-Viable Tissue

After postprocessing 123 datasets with the following algorithms (Table 2), the ISP calculated non-viable tissue (core) volumes in 115 datasets (93.5%), Cercare AI in 70 datasets (56.9%), and Cercare Threshold in 54 datasets (43.9%). The mean values revealed a highly significant deviation (*p* < 0.001) that was calculated using the Welch ANOVA. The measured infarct volumes in the follow-up CTs did not significantly differ in the post hoc analysis from the calculated core volumes of Cercare AI (*p* = 0.447) and ISP (*p* = 0.949). The mean values of the ISP core volumes and the final infarct volumes deviated the least of all, with a sensitivity of 90.2%; however, in 63.4%, a false positive core was calculated by the ISP. This led to a very bad specificity of 4.9% (Table 2)

Core measurements of the Cercare Threshold significantly deviated from the final infarct volume (*p* = 0.045), underestimating the resulting ischemic infarct. Nevertheless, Cercare Threshold reached the highest specificity with 40.2% for core volume detection. The mean core values of the ISP and Cercare Threshold deviated highly significantly (*p* < 0.001). Cercare AI reached the highest negative predictive value (NPV) with 86.6%, which was followed by the Cercare Threshold with 78.3%. All algorithms presented low positive predictive values (PPV) < 50.0% (Table 2).

### 3.6. Tissue at Risk

The algorithms calculated tissue-at-risk (hypoperfused) volumes in 123 datasets (100.0%) by ISP, in 119 datasets (96.7%) by Cercare AI, and in 81 datasets (65.9%) by the Cercare Threshold. The mean values (Table 2) revealed a highly significant deviation (*p* < 0.001).

The post hoc analysis showed that the measured infarct volumes of the follow-up plain CT agreed the best with the calculated hypoperfused volumes of the Cercare Threshold (*p* = 0.968). The volumes of Cercare AI (*p* < 0.001) and ISP (*p* < 0.001) differed significantly from the final ischemic infarct size. Figure 1 visualizes the overestimation of the tissue at risk for both algorithms. Eighty-two follow-up CTs did not show new ischemic infarcts. Only forty-two perfusion datasets were recognized as having no tissue at risk by the Cercare Threshold, and Cercare AI identified four. The ISP always calculated a hypoperfused volume. In 45 datasets, these hypoperfused volumes were located close to the skull base or cap, simply being artefacts (mean: 19.55 mL, range: 0.8–91.3 mL).

All algorithms calculated the hypoperfused and core volumes in cases of acute intracranial hemorrhage, leading to low specificity.

Cercare AI and ISP revealed a sensitivity of 100.0% without any specificity (4.9% and 0.0%). The Cercare Threshold was the most accurate algorithm in detecting tissue at risk of hypoperfusion in DCI, with a sensitivity of 78.0%, specificity of 40.2%, a PPV of 39.5%, and a NPV of 78.6%.

### 3.7. Treatment-Related Tissue Prediction

All patients received standard DCI treatment with Nimodipine orally (6 × 60 mg/day) or via gastric tube. Fifty-six CTP examinations were additionally followed by intra-arterial spasmolysis or induced hypertension. Sixteen examinations could not be grouped into “treatment” or “no treatment” due to prior intra-arterial spasmolysis being excluded for further investigations. Fifty-one CTP examinations remained without further DCI-related treatment. The distribution of the mean values for the core and hypoperfused volumes compared to the final ischemic infarct volumes is displayed in Figure 2. The four-category distribution is shown for DCI-related therapy in Appendix A and for no therapy in Appendix A.

### 3.8. DCI-Related Treatment

In the treatment group (*n* = 56), the Welch ANOVA calculated a significant deviation of the mean values for core volumes (*p* = 0.009) and a highly significant deviation for the hypoperfused volumes (*p* < 0.001).

The core volumes that were calculated by the Cercare Threshold (*p* = 1.000), Cercare AI (*p* = 0.260), and ISP (*p* = 0.031) to predict non-viable tissue better corresponded to the final ischemic infarct volume if a DCI treatment was induced. The resulting infarct volumes were smaller compared to the group without treatment (Figure 2), verifying treatment success. Therefore, the calculated hypoperfused volumes significantly deviated from the infarct volumes for all algorithms (Cercare Threshold: *p* = 0.014, Cercare AI: *p* < 0.001, ISP: *p* < 0.001).

Cercare AI performed the best in predicting already non-viable tissue, with a sensitivity of 73.7% and a specificity of 51.4% (Appendix A). Considering the prediction of infarct development, the sensitivity increased for the Cercare Threshold in the prediction of tissue at risk, with 85.7%. The specificity of Cercare AI and ISP remained low despite a 100% sensitivity.

### 3.9. No DCI-Related Treatment

In the group without treatment (*n* = 51), the means for both core and hypoperfused volumes (*p* < 0.001) deviated highly significantly from the final ischemic infarct. Without treatment, the final infarct exceeded the predicted core volumes, underlining the need for DCI-specific treatment (Figure 2). The calculated hypoperfused volumes now better corresponded to the final infarct size (Figure 2). There was no significant deviation of the infarct volume and the calculated core (Cercare Threshold: *p* = 0.057, Cercare AI: *p*= 0.156, ISP: 0.280) or hypoperfused volumes (Cercare Threshold: *p* = 0.424, Cercare AI: *p* = 0.937, ISP: *p* = 0.569) of the algorithms due to the small dataset. Only the volumes of the Cercare Threshold and ISP had significant intergroup variance (*p* < 0.001) for both perfusion volumes.

Without DCI-related treatment, the sensitivity of core prediction rose for Cercare AI (87.5%) and ISP (93.8%), with a specificity of 62.9% for Cercare AI (Appendix A). ISP presented an alarming low specificity in the prediction of non-viable tissue (8.6%) and tissue at risk (0.0%). The Cercare Threshold presented, overall, the best specificity, with 77.1% for non-viable tissue and 51.4% for tissue at risk of hypoperfusion.

### 3.10. Case 1

A 71-year-old female patient presented with an acute-onset severe headache, impaired consciousness (WFNS grade 4), and right-sided hemiparesis. Their initial MRI revealed a SAH, classified as Fisher score 3 due to ruptured aneurysms of the left PICA, accompanied by obstructive hydrocephalus. A right MCA aneurysm was incidentally found (Figure 3).

Neuroradiological coiling of both aneurysms and the placement of an external ventricular drain (EVD) were performed on the day of admission. Following stabilization, the patient was extubated without focal neurological deficits.

On day 9 post-SAH, the patient developed dysphasia and was readmitted to the intensive care unit (ICU). CT angiography demonstrated severe narrowing of the right MCA branches, which was consistent with vasospasm. CT perfusion displayed tissue at risk in the right MCA territory by ISP and Cercare AI (Figure 4), but not by the Cercare Threshold. Management of DCI development was initiated with induced hypertension, and a ventriculoperitoneal shunt was placed for ongoing hydrocephalus. Under this treatment regimen, the patient’s aphasia gradually improved.

At six months post-SAH, the patient had made a significant neurological recovery, achieving a modified Rankin Scale (mRS) score of 1, indicating no significant disability.

### 3.11. Case 2

A 58-year-old female patient was transferred to our ICU after presenting with severe headache and recurrent vomiting (WFNS grade 4). Their initial neuroimaging confirmed a SAH, classified as Fisher 3, secondary to the rupture of a left MCA aneurysm. Upon transfer, the patient underwent placement of an EVD followed by sedation and mechanical ventilation in the ICU. The following day, the patient underwent surgical clipping of the aneurysm via craniotomy.

On day 13 post-SAH, the patient developed new-onset dysphasia. Multimodal CT revealed a vasospasm in the left MCA territory. Only the Cercare AI algorithms calculated the tissue at risk of hypoperfusion in the left MCA territory (Figure 5). ISP displayed artefacts close to the EVD. In response, induced hypertension was implemented as a treatment option. Subsequently, the dysphasia resolved and no additional focal neurological deficits were observed.

At six months post-SAH, the patient achieved a mRS of 1, indicating no significant disability.

## 4. Discussion

In diagnosing DCI after a SAH, CT perfusion is an equally important part of a multimodal CT examination, as a neurological assessment of the patient may sometimes be impossible. In particular, unconscious patients are challenging to monitor. DCI, as defined by Vergouwen et al., includes both clinical deterioration and cerebral infarction [14]. DCI is more complex than simply being a result of arterial narrowing. A multifactorial process involving blood–brain barrier disruption, microthrombosis, cortical spreading depolarization, and a loss of cerebral autoregulation leads to DCI. While infarction is irreversible, functional deterioration resulting from cerebral hypoperfusion can often be reversed with appropriate treatment, such as induced hypertension or intra-arterial therapies, emphasizing the importance of early detection and intervention in patients with DCI. TCD and CTP are both monitoring methods with a moderate level (2a) of evidence according to the 2023 “Guidelines for the Management of Patients with Aneurysmal Subarachnoid Hemorrhage” of the American Heart Association and the American Stroke Association [3]. TCD is a non-invasive bedside monitoring that is strongly limited by operator dependence. Compared with CTA, the sensitivity and specificity for detecting vasospasm in the MCA are rather high (~89–98%) [15]. CTP provides a functional diagnosis of cerebral circulation, allowing for the assessment of ischemia arising from thromboembolism, or micro- or macrovasospasm. Operator independence drives its complementary value.

Commonly used CTP thresholds include a rCBF < 30% for the core volume and a Tmax >6 s for hypoperfused tissue [16,17]. These thresholds were specifically designed for acute ischemic stroke. They slightly underestimate the ischemic core infarct volume, thereby favoring inclusion for treatment [18,19]. As highlighted by Chung et al. [20], the easy accessibility of CTP images overshadows the complexity of the underlying computational postprocessing workflow. It may lead to overconfidence in interpretation, thereby favoring diagnostic errors. As highlighted by Demeestere et al. [21], the infarcted tissue will not be identified as non-viable tissue on CTP if the CBF exceeds the threshold for ischemic core detection. The inspection of the plain CT for subacute or established infarct development is thus mandatory, especially in patients with SAH. CTP has a relatively limited spatial resolution and, consequently, a limited sensitivity in detecting lacunar or small subcortical infarcts. This resulted in a sensitivity of only 62% compared to diffusion-weighted magnetic resonance imaging in another study [22]. Sensitivity for infratentorial lesions is, however, low.

In our study, three algorithms with divergent thresholds were compared to identify the best-fitting algorithm for DCI. Previous studies evaluated arrival times [23] and different thresholds [24] for ISP in predicting final infarct sizes in ischemic stroke. A poor volumetric agreement was found for all methods in patients with complete recanalization. Koopman et al. [25] compared ISP, RAPID, and syngo.via being superior in categorizing patients with small core infarcts. To our knowledge, there is no study published comparing Cercare Threshold or Cercare AI to other software packages.

The Cercare Threshold algorithm, applying the commonly used CT perfusion thresholds, derived the highest specificity for hypoperfused and core volumes compared to the final infarct volumes in follow-up examinations. It shows tissue that is highly endangered of experiencing hypoperfusion. The AI method calculated a larger hypoperfused volume than the threshold method with a Tmax > 6 s. One of the main reasons is that the AI model is based on TTP outlines (created by clinical experts), along with other maps, and not the Tmax, as is in the threshold method. TTP lesions are larger than the Tmax lesions and, thus, correspond more to all affected tissue. At the same time, TTP is more susceptible to a low cardiac ejection fraction, proximal stenosis, and poor bolus injections. In the treatment-based approach, core volumes of all algorithms corresponded more to the final infarct volumes, underlining the success of intra-arterial spasmolysis and induced hypertension in preventing infarct growth. The hypoperfused volumes did not become ischemic infarcts and were, therefore, overestimated. Rava et al. [26] described the same phenomenon for patients undergoing a thrombectomy or not.

Correct prediction of ischemic core and penumbra in strokes was crucial for patient selection before the randomized controlled trials investigated large core infarcts for mechanical thrombectomy [27]. With CTP being applied in different clinical scenarios, physicians must understand that using different software packages may produce different results that can impact their decisions. In DCI, there is no clear threshold indicating whether treatment is necessary or not [28]. Lolli et al. reviewed “early” (<72 h from aSAH) and “late” (>72 h from aSAH) CTP imaging, with an aSAH being more informative when performed in a later time window. Lower tissue-at-risk thresholds (i.e., Tmax > 4 s) were observed in an aSAH compared to an acute ischemic stroke. Similarly, the importance of CTP on admission as an individual baseline reference for subsequent CTP studies was emphasized. Automated measurements from voxelwise analyses presented major limitations in an aSAH due to the high incidence of DCI-unrelated parenchymal damage. All algorithms in our analysis included an acute ICH in the calculated hypoperfused and core volumes. Furthermore, intracranial devices are responsible for severe artefacts that render automated measurements.

If no DCI-related treatment was performed, the core volumes were underestimated because tissue at risk of hypoperfusion progressed to ischemia. DCI is a progressive disease that can aggravate. Different from vessel occlusion in ischemic stroke, the absence of a clear perfusion deficit does not necessarily mean there is no tissue at risk. Sanelli et al. [29] found that aSAH patients with >50% angiographic narrowing of a given artery tend to have a perfusion deficit with reduced CBF. Contradictory results reported that patients with large or medium-sized arteries had vasospasm without having any perfusion deficit [30,31]. In the dynamic nature of vasospasm, it can regress in one territory following selective treatment on one CTP scan while, at the same time, affecting a new territory, which is difficult to quantify in a method that is based on a side-by-side comparison in bilateral pathology. Beyond that, cortical spreading depolarization and seizures are accompanied by transitory hyperperfusion [12], which is not evaluated by the existing software packages. The evaluated three algorithms showed an incredibly high sensitivity of Cercare AI and ISP in detecting hypoperfused tissue, with an extremely poor specificity of <5%. These findings indicate limited evidence for clinical use of Cercare AI and ISP to predict tissue at risk of DCI. Cercare AI is superior in predicting core volumes independent of treatment. The Cercare Threshold performed, overall, the best by applying the commonly used ischemic stroke thresholds. Thirty-one patients underwent two or more CT perfusion examinations. The first CT perfusion often revealed slight changes in the border zones, showing increased OEF and MTT, which aggravated territorial hypoperfusion, as presented in cases 1 and 2. The AI method overestimated tissue at risk of hypoperfusion by detecting early elevations in MTT, TTP, CTH, and OEF. Previous studies highlighted the importance of MTT as a radiological surrogate for DCI [32,33]. Patients with unfavorable outcomes (mRS ≥ 2) experienced significantly earlier MTTPEAK onsets in serial CT perfusion measurements than patients with favorable outcomes (mRS 0–1) [32] due to microvascular disturbances. Overestimation of hypoperfused tissue can lead to earlier treatment in the best-case scenario; however, the treatment options are limited and must be carefully chosen. Therapy directed at vasospasm alone has had a limited impact on DCI or mortality in patients with an aSAH. Induced hypertension, intra-arterial vasodilator therapy, and cerebral angioplasty are treatment options with a weak level (2b) of evidence [3]. Nimodipine and Milrinone are the most commonly applied drugs for intra-arterial spasmolysis [34,35]. Both drugs influence the amount of cytosolic calcium in different ways. Systemic side effects like hypotension, tachycardia, and electrolyte disturbances limit the effectiveness of aggressive treatment [36,37]. Notably, it is crucial to recognize that CTP evaluates the hemodynamic status of the brain at a particular time and does not reflect actual tissue viability. Moreover, shorter periods of severe hypoperfusion are less likely to result in permanent injury. CTP maps are static snapshots of the hemodynamic tissue state (hypoperfusion severity) rather than the actual tissue fate (infarction or no infarction). An extensive but individualized collateral circulation system secures the blood supply to the brain. The extent of collateral flow varies significantly between individuals. The degree of collateral supply through peripheral leptomeningeal sources correlates with the presence of a smaller final infarct volume in ischemic stroke [38]. The CTA scores that are used to evaluate the collateral status are based on contrast filling of the MCA territory [39] or by cortical vein opacification [40]. In intracranial atherosclerotic disease (ICAD) or vasospasm, assessing the collateral supply is even more challenging. Without a proximal vessel occlusion, contrast is visible in the whole MCA territory and the cortical veins, even though the blood flow is delayed. Digital subtraction angiography (DSA) grading systems are only applicable if invasive endovascular treatment is performed [41]. CTP is the only non-invasive monitoring that gives information on time parameters of the affected and adjacent territories. Hyperperfusion in the vertebrobasilar system might be a sign of extensive collateral supply.

According to previous DSA-based investigations [41,42], patients with severe vasospasm are more likely to develop collaterals. Al-Mufti et al. [42] suggested that collateral grades did not affect DCI development. In contrast, patients showed poor outcomes with poor collateral status in the SAH cohort of Topcu et al. [41]. Moftakhar et al. [43] studied cerebral vasospasm in SAH in childhood. They found that children with robust cerebral collateral blood flow rarely developed neurological symptoms due to cerebral vasospasm. In our retrospective analysis, DCI occurred with induced hypertension (33.3%), after intra-arterial spasmolysis (34.4%), and without DCI-specific treatment (35.3%), without clearly favoring one treatment option. DCI-specific treatment could prevent volumetric infarct growth. Especially in unconscious patients, we need a sensitive screening method to detect early patients at risk of developing DCI for staged treatment options. By considering the perfusion status of the bordering territories, especially the vertebrobasilar system, high-risk and low-risk patients can be selected. Invasive treatment options often accompany systemic side effects that can impact the patient’s clinical outcome. A new CT perfusion algorithm that displays changes in CBF compared to the reference values may be able to identify hypo- and hyperperfused areas of cerebral tissue. Non-invasive measurements of the collateral status can increase the specificity of identifying patients at a real risk of developing DCI without DCI-specific treatment, whereas drug-induced side effects could be diminished otherwise.

### Limitations

The minimum scan interval was limited by the desired anatomical coverage of 8 cm versus the axial coverage provided by the scanner of 4 cm. To increase the coverage with CT perfusion during a single bolus of contrast material, Roberts et al. [44] developed a cine-mode technique, the toggling-table technique (also known as the step-and-shooter shuttle mode), in which the table is moved between two locations during the bolus. Due to restrictions on table movement, the minimal temporal sampling interval for our CT scanner was 4 s. According to the calculations of Wintermark et al. [45], this technique results in significant differences in the calculation of CBV and CBF compared to their reference standard (3 s optimum).

The automated selection of AIF and VOF for the Cercare algorithms increases the interrater reliability. Whereas these basic functions need to be manually selected in the Intellispace Portal software. Determining the terminal segment of the ICA (AIF) and superior sagittal sinus (VOF) as a target was the applied rule, which can vary between raters in clinical practice. In this study, only one neuroradiologist generated the analysed perfusion maps by applying the same rules overall.

Plain CT scans were exclusively segmented via the 3D slicer to define the final infarct volumes, owing to clinical practice. MRI scan capacity has to be distributed among several competing departments. High-field MRI has additional limitations, with the main ones being susceptibility to ferromagnetic material or contraindications due to the equipment needed by an intensive care unit patient during image acquisition. Therefore, the radiological monitoring was realized via multimodal CT to address the question of DCI. Cortical and small-volume ischemia may be missed in DCI detection in this approach. This limits the precision of the ground truth against which the CTP performance is measured. Additionally, suboptimal scan quality can lead to interrater variability. The 3D slicer is a user-friendly, open-source software that provides thresholding for segmentation, but needs visual control and adjustment. This study lacks the evaluation of spatial resolution, e.g., the Dice similarity coefficient. A comparison of spatial resolution requires the generation of 3D contours for the co-registration of infarct volumes and calculated hypoperfused and core volumes. The same volume measurements can be acquired in different brain regions without any spatial overlap. Our analysis cannot precisely differentiate between artefact detection (e.g., skull base), correct infarct prediction, and new infarct development outside the perfusion deficit.

We did not perform a Receiver Operating Characteristics (ROC) analysis because of a missing graduated thresholding factor (e.g., Tmax). Two software tools, each with its own generated perfusion maps, were evaluated. Each of these used divergent AIF, VOF, and artefact corrections. The three algorithms that were used to calculate the hypoperfused and core volumes had divergent underlying thresholding parameters (ISP: rMTT and CBV, Cercare Threshold and AI: rCBF and Tmax). These preconditions are inappropriate for performing a ROC analysis. Moreover, binary classifiers are statistical and computational models that divide a dataset into two groups: positives and negatives. ROC’s primary limitation is that the ROC curve can be skewed when the classes in a study are imbalanced. Two of the three algorithms clearly overestimated the hypoperfused volumes, resulting in low specificity. The visual interpretability of ROC curves in the context of imbalanced datasets can be deceptive regarding conclusions about the reliability of classification performance due to an intuitive but incorrect interpretation of specificity.

Philips updated their workspace (Advanced Visualization Workspace) and CT perfusion threshold values for ischemic stroke analysis to the commonly applied ones (rCBF < 30% for the core volume and a Tmax > 6 s for hypoperfused tissue), which are similar to the Cercare Threshold methods.

## 5. Conclusions

Cercare AI and ISP presented high sensitivity in detecting perfusion deficits but demonstrated an alarming poor specificity in predicting DCI. Cercare Threshold applied the commonly used thresholds for ischemic strokes, and was superior for tissue-at-risk detection in DCI. The highest obtained specificity of 51.4% without DCI-related treatment underlines the need for DCI-specific thresholds based on reference values. Side comparisons in a bilateral pathology are prone to error. The collateral status needs to be taken into account to distinguish between high- and low-risk patients of developing DCI.

## Figures and Tables

**Figure 1 diagnostics-15-02236-f001:**
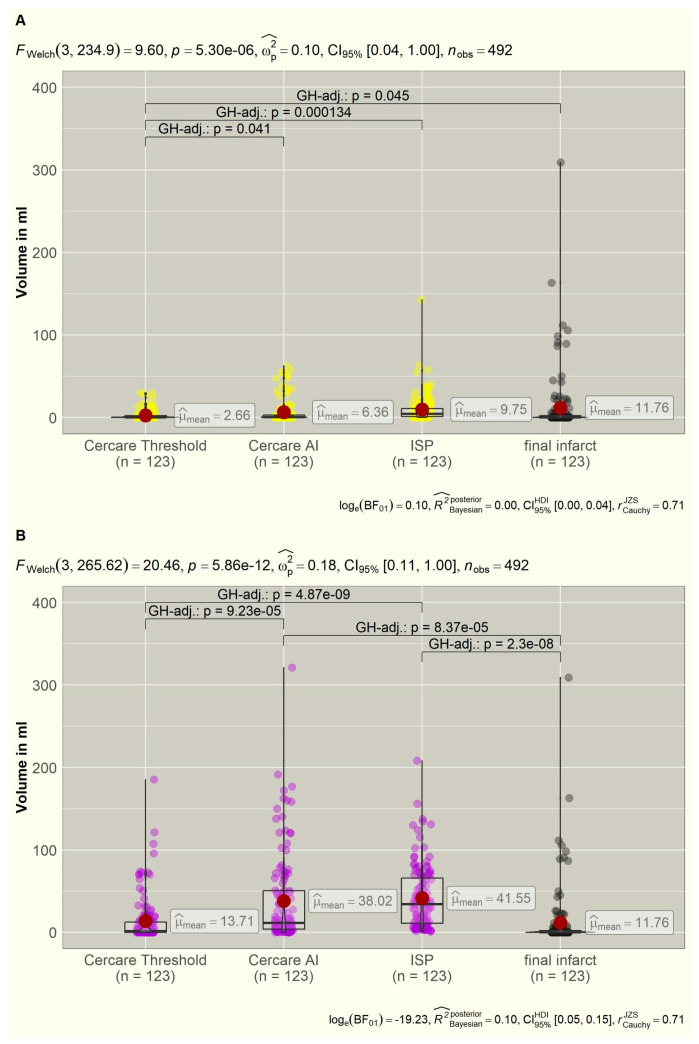
(**A**,**B**): Distribution of the calculated core (yellow) and hypoperfused (violet) volumes of the three algorithms compared to the final ischemic infarct volumes (grey). Significant intergroup variations are displayed in brackets with *p*-values. AI = artificial intelligence, FWelch = Welch ANOVA, GH = Games–Howell, ISP = Intellispace Portal.

**Figure 2 diagnostics-15-02236-f002:**
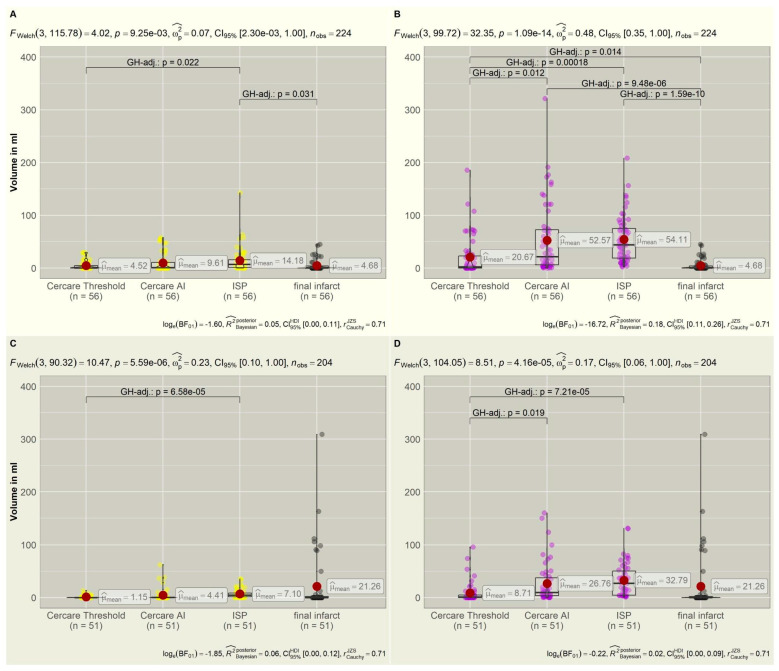
(**A**,**B**): The top row shows the distribution of core (yellow) and hypoperfused (violet) volumes compared to the final infarct volumes (gray) if CTP was followed by intra-arterial spasmolysis or induced hypertension. DCI-specific treatment could reduce infarct progression. The final infarct volumes correspond to the non-viable tissue measured as the core volume in the CTP. (**C**,**D**): The bottom row displays the distribution of core (yellow) and hypoperfused (violet) volumes in correlation to the final infarct volumes (gray) without DCI-related treatment. Infarct volumes are larger without treatment compared to the treatment group in the top row, approximating the hypoperfused volumes. Significant intergroup variations are displayed in brackets with *p*-values. CTP = computer tomography perfusion, DCI = delayed cerebral ischemia, F_Welch_ = Welch ANOVA, GH = Games–Howell.

**Figure 3 diagnostics-15-02236-f003:**
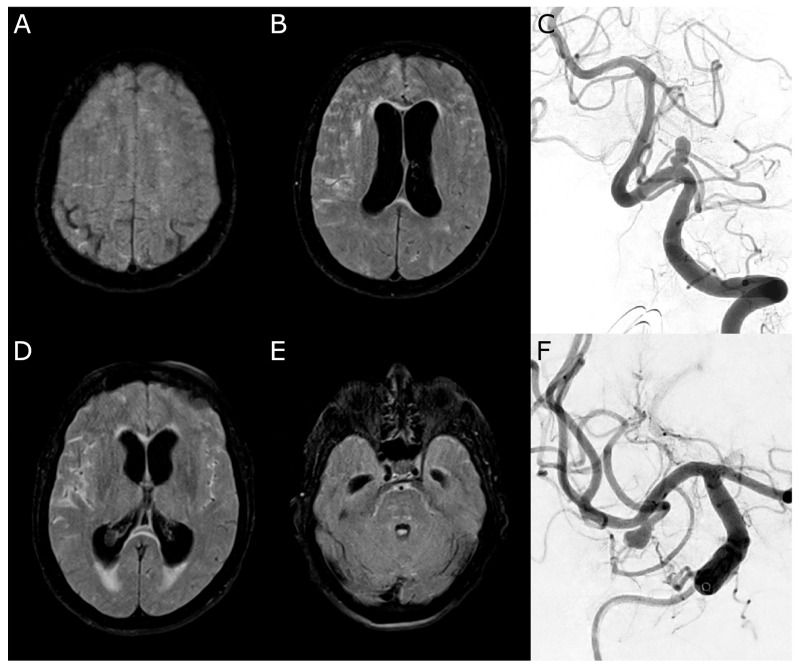
Case 1. (**A**,**B**,**D**,**E**): Initial magnetic resonance imaging (Fluid Attenuated Inversion Recovery) with a subarachnoid hemorrhage more affecting the right side and acute hydrocephalus. (**C**): Digital subtraction angiography confirming an irregularly shaped aneurysm of the left posterior inferior cerebellar artery. (**F**): Incidental finding of a right-sided middle cerebral artery aneurysm.

**Figure 4 diagnostics-15-02236-f004:**
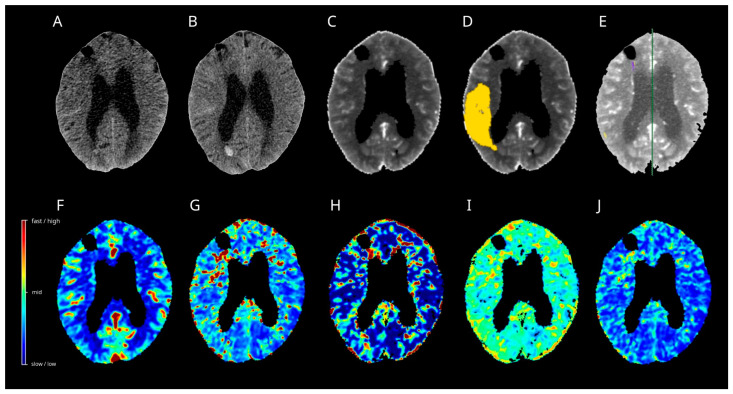
Case 1. Multimodal CT on day 9 after SAH (**A**,**C**–**J**) and follow-up plain CT on day 11 after SAH (**B**) are presented. The artificial intelligence algorithm (**D**) calculated a huge hypoperfused area in the right MCA territory (yellow), with increases in the time to maximum (**H**) and a decrease in the oxygen extraction fraction (**I**). The remaining perfusion maps, regional cerebral blood flow (**F**), mean transit time (**G**), and capillary transit time heterogeneity (**J**) did not display noticeable changes. The Threshold (**C**) and Intellispace Portal (**E**) algorithms calculated no tissue at risk in the right MCA territory-no cerebral ischemia developed in the follow-up CT (**B**). CT = computer tomograph, MCA = middle cerebral artery, SAH = subarachnoid hemorrhage.

**Figure 5 diagnostics-15-02236-f005:**
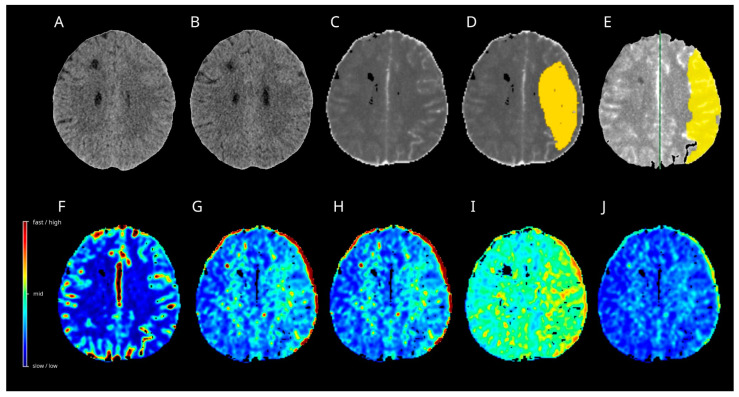
Case 2. Multimodal CT on day 13 after SAH (**A**,**C**–**J**) and follow-up plain CT one month after SAH (**B**) are shown. The algorithm of Artificial Intelligence (**D**) and Intellispace Portal (**E**) calculated a huge hypoperfused area in the left MCA territory (yellow), with increases in the mean transit time (**G**) and time to maximum (**H**). The oxygen extraction fraction (**I**) displayed an increase. Capillary transit time heterogeneity (**J**) showed only slight increases in the border zones, while a relative cerebral flow did not display noticeable changes in the left MCA territory. No cerebral ischemia developed in the follow-up CT (**B**). CT = computer tomograph, MCA = middle cerebral artery, SAH = subarachnoid hemorrhage.

**Table 1 diagnostics-15-02236-t001:** Demographics of patients suffering from aneurysmal subarachnoid hemorrhage.

Clinical characteristics on admission		
Patients		75 (100.0)
Female (%)		55 (73.3)
Mean Age (range)		56 (32–90)
Mean BMI (range)		27 (18–40)
Fisher score (%)	1–3	29 (38.7)
	4	46 (61.3)
WFNS grading (%)	1–3	44 (58.7)
	4–5	31 (41.3)
Location of intracranial aneurysm		
AComA (%)		23 (30.6)
PComA (%)		8 (10.7)
MCA Bifurcation (%)		20 (26.7)
Pericallosal artery (%)		5 (6.7)
ICA (%)		8 (10.7)
BA (%)		5 (6.7)
PICA (%)		1 (1.3)
VA (%)		1 (1.3)
Multiple (%)		4 (5.3)
Aneurysm treatment		
Timing of aneurysm treatment in days (median, range)		0.00 (0–19)
Coiling (%)		43 (57.3)
Clipping (%)		19 (25.3)
Coiling + Clipping (%)		4 (5.3)
Flow diverter (%)		4 (5.3)
Coiling + Flow diverter (%)		2 (2.7)
Coiling + Contour device		1 (1.3)
WEB (%)		2 (2.7)
Delayed cerebral ischemia (DCI)		
Clinical deterioration		53 (70.7)
DCI-related infarction		32 (42.7)
DCI treatment		
Induced hypertension (%)		13 (17.3)
Intra-arterial spasmolysis (%)		29 (38.7)
None (%)		33 (44.0)
Outcome		
mRS after 6 months (%)	0–3	39 (52.0)
	4–6	36 (48.0)

AComA = anterior communicating artery, BA = basilar artery, BMI = body mass index, DCI = delayed cerebral ischemia, ICA = internal carotid artery, MCA = middle cerebral artery, mRS = modified RankinScale, PComA = posterior communication artery, PICA = posterior inferior cerebellar artery, VA = vertebral artery, WFNS = World Federation of Neurosurgical Societies, WEB = Woven EndoBridge.

**Table 2 diagnostics-15-02236-t002:** Perfusion deficit correlated with final infarct development.

	Volume		Cercare Threshold	Cercare AI	ISP
Overall		n (%)	123 (100.0)	123 (100.0)	123 (100.0)
	Hypoperfused	mean (range)	13.7 mL (0.0–185.5 mL)	38.0 mL (0.0–321.2 mL)	41.6 mL (0.8–208.4 mL)
	Core	mean (range)	2.7 mL (0.0–29.5 mL)	6.4 mL (0.0–62.4 mL)	9.8 mL (0.0–142.9 mL)
No abnormality		n (%)	33 (26.8)	4 (3.3)	0 (0.0)
	Hypoperfused	mean (range)	0 (0.0)		
		n (%)	54 (43.9)	46 (37.4)	4 (3.3)
	Core	mean (range)	0 (0.0)		
Underestimated progressive infarct		n (%)	9 (7.3)	0 (0.0)	0 (0.0)
	Hypoperfused	mean (range)	0 (0.0)		
		n (%)	15 (12.2)	7 (5.7)	4 (3.3)
	Core	mean (range)	0 (0.0)		
Reversible perfusion deficit		n (%)	49 (39.8)	78 (63.4)	82 (66.7)
	Hypoperfused	mean (range)	16.4 mL (0.1–185.5 mL)	30.0 mL (0.1–321.2 mL)	34.2 mL (0.8–156.2 mL)
		n (%)	28 (22.8)	36 (29.3)	78 (63.4)
	Core	mean in mL(range)	6.4 mL (0.1–29.3 mL)	12.0 mL (0.1–57.9 mL)	8.1 mL (0.1–40.0 mL)
Progressive infarct		n (%)	32 (26.0)	41 (33.3)	41 (33.3)
	Hypoperfused	mean (range)	27.6 mL (0.3–121.5 mL)	57.0 mL (0.2–191.4 mL)	56.3 mL (2.1–208.4 mL)
		n (%)	26 (21.1)	34 (27.6)	37 (30.1)
	Core	mean (range)	5.6 mL (0.1–29.5 mL)	10.3 mL (0.1–62.4 mL)	15.3 mL 0.3–142.9 mL
Sensitivity	Hypoperfused		78.0%	100.0%	100.0%
Specificity			40.2%	4.9%	0.0%
PPV			39.5%	34.5%	33.3%
NPV			78.6%	100.0%	/
Sensitivity	Core		63.4%	82.9%	90.2%
Specificity			65.9%	56.1%	4.9%
PPV			48.1%	48.6%	32.2%
NPV			78.3%	86.8%	50.0%

AI = artificial intelligence, CT = computer tomography, ISP = Intellispace Portal, NPV = negative predictive value, PPV = positive predictive value.

## Data Availability

The datasets generated and analyzed during the current study are available in the Zenodo repository (https://www.zenodo.org (accessed on 28 June 2025)) digital object identifier number: 10.5281/zenodo.15761670; further inquiries can be directed to the corresponding author.

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
