# Peer review of "A Comparison of Three Perfusion Algorithms in Patients at Risk of Delayed Cerebral Ischemia After Subarachnoid Hemorrhage"

_diagnostics, 2025, doi:10.3390/diagnostics15172236_

Round 1

Reviewer 1 Report

Comments and Suggestions for Authors

The paper by Falter and colleagues presents the report of the study compating three algorithms to the perfusion CT data analysis for the detection of delayed cerebral ischemia in SAH patients.  The authors demonstrated that only one approach provided agreement with the follow-up infarct volume; in general, algorithms had rather low specificity (0–36.7%) and very high variability. The authors conclude that new algorithm for the analysis of CTP data is needed that might provide more reliable results.

The  paper is clear and presented in a well-structured manner. The authors analyized 132 CTP datasets from 85 SAH patients using appropriate methods. The results are sound, they are illustated with two clicnical cases. The Discussion section covers the problem, and it might be improved by including previous papers that compare vaaious CTP data analysis approaches. The conclusions  are consistent with the results.

Author Response

Response to Reviewer Comments 1

The paper by Falter and colleagues presents the report of the study compating three algorithms to the perfusion CT data analysis for the detection of delayed cerebral ischemia in SAH patients.  The authors demonstrated that only one approach provided agreement with the follow-up infarct volume; in general, algorithms had rather low specificity (0–36.7%) and very high variability. The authors conclude that new algorithm for the analysis of CTP data is needed that might provide more reliable results.

The  paper is clear and presented in a well-structured manner. The authors analyized 132 CTP datasets from 85 SAH patients using appropriate methods. The results are sound, they are illustated with two clicnical cases.

The Discussion section covers the problem, and it might be improved by including previous papers that compare vaaious CTP data analysis approaches. The conclusions  are consistent with the results.

Answer:

Thank you very much for this esteemed review. We extended the discussion and included previous papers that compare various CTP data analysis approaches, as follows:

“Commonly used CT perfusion thresholds include rCBF <30% for core volume and Tmax >6s for hypoperfused tissue [16,17]. These thresholds were specifically designed for acute ischemic stroke. They slightly underestimate the ischemic core infarct volume, thereby favoring inclusion for treatment [18,19]. As highlighted by Chung et al. [20], the easy accessibility of CTP images overshadows the complexity of the underlying computational postprocessing workflow. It may lead to overconfidence in interpretation, thereby favoring diagnostic errors.

In our study, three algorithms with divergent thresholds were compared to identify the best-fitting algorithm for DCI. Previous studies evaluated arrival times [21] and different thresholds [22] for ISP in predicting final infarct size in ischemic stroke. A poor volumetric agreement was found for all methods in patients with complete re-canalization. Koopman et al. [23] compared ISP, RAPID, and syngo.via being superior in categorizing patients with small core infarcts. To our knowledge, there is no study published comparing Cercare Threshold or Cercare AI to other software packages.

The Cercare Threshold algorithm, applying the commonly used CT perfusion thresholds, derived the highest specificity for hypoperfused and core volumes compared to the final infarct volumes in follow-up examinations. It shows tissue highly endangered of hypoperfusion. The AI method calculated a larger hypoperfused volume than the Threshold method with Tmax >6s. One of the main reasons is that the AI model is based on TTP outlines (created by clinical experts), along with other maps, and not Tmax, as in the Threshold method. TTP lesions are larger than the Tmax lesions and thus correspond more to all affected tissue. At the same time, TTP is more susceptible to low cardiac ejection fraction, proximal stenosis, and poor bolus injections. In the treatment-based approach, core volumes of all algorithms corresponded more to the final infarct volumes, underlining the success of intra-arterial spasmolysis and induced hypertension in preventing infarct growth. The hypoperfused volumes did not become ischemic infarcts and were therefore overestimated. Rava et al. [24] described the same phenomenon for patients undergoing thrombectomy or not.

Correct prediction of ischemic core and penumbra in stroke was crucial for patient selection before randomized controlled trials investigated large core infarcts for mechanical thrombectomy [25]. With CTP being applied in different clinical scenarios, physicians must understand that using different software packages may produce different results that can impact their decisions. In DCI, there is no clear threshold indicating whether treatment is necessary or not [26]. Lolli et al. reviewed “early” (<72 h from aSAH) and “late” (>72 h from aSAH) CTP imaging, with aSAH being more informative when performed in a later time window. Lower tissue at risk thresholds (i.e., Tmax > 4s) were supposed in aSAH compared to acute ischemic stroke. Likewise, the importance of CTP on admission as an individual baseline reference for subsequent CTP studies was emphasized. Automated measurements from voxelwise analyses presented major limitations in aSAH due to the high incidence of DCI-unrelated parenchymal damage. All algorithms in our analysis included acute ICH in the calculated hypoperfused and volumes. Furthermore, intracranial devices are responsible for severe artefacts that render automated measurements.

 If no DCI-related treatment was performed, the core volumes were underestimated because tissue at risk of hypoperfusion progressed to ischemia. DCI is a progressive disease that can aggravate. Different from vessel occlusion in ischemic stroke, the absence of a clear perfusion deficit does not necessarily mean there is no tissue at risk. Sanelli et al. [27] found that aSAH patients with >50% angiographic narrowing of a given artery tend to have a perfusion deficit with reduced CBF. Contradictory results reported that patients with large or medium-sized arteries had vasospasm without having any perfusion deficit [28,29]. In the dynamic nature of vasospasm, it can regress in one territory following selective treatment on one CTP scan while at the same time affecting a new territory, which is difficult to quantify in a method that is based on a side-by-side comparison in a bilateral pathology. Beyond that, cortical spreading depolarization and seizures are accompanied by transitory hyperperfusion [12], which is not evaluated by the existing software packages. The evaluated three algorithms showed an incredibly high sensitivity of Cercare AI and ISP in detecting hypoperfused tissue, with an extremely poor specificity of <5%. These findings indicate limited evidence for clinical use of Cercare AI and ISP to predict tissue at risk of DCI. Cercare AI is superior in predicting core volumes independent of treatment. Cercare Threshold performed overall the best by applying the commonly used ischemic stroke thresholds.”

Reviewer 2 Report

Comments and Suggestions for Authors

The authors assessed the volumetric agreement of the CTP ischemic core and hypoperfused tissue in patients with SAH using three different CTP stroke algorithms: (1) Brain Perfusion of Intellispace Portal, (2) Cercare Threshold  and (3) Cercare Artificial Intelligence. Although unfortunately not clearly articulated in the Abstract or the Introduction, the rationale for doing this comparison is that commonly used CT perfusion thresholds include rCBF <30% for core volume and Tmax >6s for hypoperfused tissue detection, which are thresholds specifically developed for acute ischemic stroke, that may or may not apply to SAH and vasospasm/delayed ischemia.

CRITIQUE

TITLE. The title is misleading and uninformative. I suggest instead: “A comparison of three CT perfusion algorithms in patients at risk for cerebral vasospasm”

ABSTRACT – it is not clear what standard was used to compare the 3 algorithms. “CT infarct volumes in plane CTs” is mentioned, but it’s not clear if this is what was used as the standard to compare the 3 algorithms against. Also, “plane” should be “plain”. The standard for comparison needs to be clearly identified and labeled as such, in both the Abstract and the Methods.

That abstract is missing a meaningful conclusion. “Low specificity of tissue-at-risk prediction of all algorithms underlines the need for new perfusion postprocessing” is not a meaningful conclusion. A more meaningful conclusion is found in the Discussion: “The Cercare Threshold algorithm, applying the commonly used CT perfusion thresholds, derived the highest specificity for hypoperfused and core volumes compared to existing and final infarct volumes.” Something like this would be more informative in the Abstract

METHODS

Methods p.5, listing numerous derived values that are not further presented or discussed or used to compare algorithms is unnecessary. Please delete all irrelevant measures that are not used for the comparison of the 3 methods.

P3, line 89: “DCI was defined as cerebral infarction” Was there correspondence between hypoperfused region of brain and neurological abnormality? Please state explicitly

How was the core volume in the plain CT quantified?

STATISTICS

Statistics – the authors indicate that a Welsh test was used, but it is not clear if this is Welsh test for 2 groups or for 3 or more groups (Welsh ANOVA). Please clarify.

When analyzing data from 3 or more groups with unequal variances, the standard ANOVA test is not appropriate. Instead, robust alternatives like Welch's ANOVA or Brown-Forsythe's test should be used. These tests do not assume equal variances and are more reliable when dealing with heteroscedasticity (unequal variances).

The authors used Dunnett's post hoc test but this test assumes equal variances across the groups being compared. The Games-Howell post hoc test, like Welch’s analysis of variance, does not require the groups to have equal standard deviations.

Section 3.3 in Results should be placed in Methods/statistics.

FIGURES AND LEGENDS

All the legends are very poorly presented, essentially unintelligible or uninformative. In the legends, the letter that identifies each panel is sometimes at the beginning, other times in the middle of the description, the ROI that is colored is not clearly associated with the panel designated by the letter, and descriptions are very brief, incomplete and overall uninformative. This is all unnecessarily confusion and annoying. Please edit all figure legends extensively to improve clarity.

Fig. 2 figure legend is poorly written /confusing. “hypoperfused volumes (yellow/green) calculated by (C) Threshold; (D) Artificial Intelligence; (E) Intellispace Portal;” Here, yellow/green seems to belong to (D), not (C). Also, why do the authors show yellow ROIs only in (D) – why not show the ROI for the 3 methods for visual comparision? This is quite confusing.

Fig 3 legend is a similarly confusing, unclear presentation referring to colored ROI in a B&W scan and not being clear about the time of the scans

Fig. 4 is not referred to in the Results section and Fig 5 is referred to only in the Discussion. What is the point of these figures? They seem entirely superfluous.

CASES

Section 3.6 case 1 “ruptured aneurysms of the left PICA and the right MCA” It is very unusual, almost unheard of, to have two aneurysm rupture at the same time in the same patient. It is much more likely that one aneurysm ruptured and the other one was found incidentally. The presentation with right-sided hemiparesis suggests that the R MCA aneurysm may not be the one that bled, but later the patient was found to have severe narrowing of the R MCA. All of this is quite atypical, most unusual. I wonder if there’s an error in this presentation. The CT at presentation to show the pattern of hemorrhage may help explain things.

In Fig. 5, showing “time to maximum” is pointless. This was not a variable that was used to compare the 3 methods, and it conveys no useful information about the algorithms. There is no explanation of why it is shown.

Author Response

Response to Reviewer 2 Comments

Comment 1:

The authors assessed the volumetric agreement of the CTP ischemic core and hypoperfused tissue in patients with SAH using three different CTP stroke algorithms: (1) Brain Perfusion of Intellispace Portal, (2) Cercare Threshold  and (3) Cercare Artificial Intelligence. Although unfortunately not clearly articulated in the Abstract or the Introduction, the rationale for doing this comparison is that commonly used CT perfusion thresholds include rCBF <30% for core volume and Tmax >6s for hypoperfused tissue detection, which are thresholds specifically developed for acute ischemic stroke, that may or may not apply to SAH and vasospasm/delayed ischemia.

Answer to Comment 1:

Thank you very much for your effort and time to improve our manuscript with your ideas and meaningful comments.

We adapted the abstract as follows:

“Commonly used CTP thresholds include rCBF <30% for core volume and Tmax >6s for hypoperfused tissue detection in acute ischemic stroke. These stroke algorithms computing thresholds compared to the contralateral hemisphere may or may not apply to detect tissue at risk of DCI.”

TITLE.

Comment 2:

The title is misleading and uninformative. I suggest instead: “A comparison of three CT perfusion algorithms in patients at risk for cerebral vasospasm”

Answer to Comment 2:

We changed the title to:

“A Comparison of Three Perfusion Algorithms in Patients at Risk of Delayed Cerebral Ischemia after Subarachnoid Hemorrhage”

Comment 3:

ABSTRACT – it is not clear what standard was used to compare the 3 algorithms. “CT infarct volumes in plane CTs” is mentioned, but it’s not clear if this is what was used as the standard to compare the 3 algorithms against. Also, “plane” should be “plain”. The standard for comparison needs to be clearly identified and labeled as such, in both the Abstract and the Methods.

Answer to Comment 3:

The standard is described as follows in the abstract (lines 22-24):

“We aimed to quantify the volumetric agreement of the CTP ischemic core and hypoperfused tissue estimated with three different stroke algorithms compared to final infarct volumes as the standard.”

The standard is described as follows in the methods (lines 198-203):

“Standard

Infarct volume at follow-up CT was used as the standard measure. The plain CT scan at the time-point of CTP (baseline) was compared to previous CT scans to exclude prior or posthemorrhagic or ischemic strokes from measurements. Then, ischemic brain lesions in baseline and follow-up CTs (following available CT scan) were delineated using a semi-automated technique (3D slicer, https://www.slicer.org/) by a neuroradiologist who was blinded to clinical data and perfusion maps. A region-of-interest thresholding with manual adjustments was used to segment hypodense lesions in high-resolution CT scans with a slice thickness of 1.5 mm or less.”

The spelling error is corrected.

 Comment 4:

That abstract is missing a meaningful conclusion. “Low specificity of tissue-at-risk prediction of all algorithms underlines the need for new perfusion postprocessing” is not a meaningful conclusion. A more meaningful conclusion is found in the Discussion: “The Cercare Threshold algorithm, applying the commonly used CT perfusion thresholds, derived the highest specificity for hypoperfused and core volumes compared to existing and final infarct volumes.” Something like this would be more informative in the Abstract

 Answer to Comment 4:

We adapted the conclusion of the abstract as follows (lines 34-36):

“The calculated core volumes corresponded best to the final infarct volumes if DCI-related treatment was performed subsequently. Additional postprocessing improved the calculation of core volumes, but overestimated tissue at risk of hypoperfusion in DCI. Whereas the accuracy of tissue-at-risk prediction accelerated without treatment, underlining the importance of intra-arterial spasmolysis and induced hypertension in the prevention of DCI. Cercare AI and ISP revealed a sensitivity of 100% each with a serious low specificity < 5% independent of treatment. Overall, Cercare Threshold, applying the commonly used stroke thresholds, performed the best in predicting tissue at risk of hypoperfusion in DCI.”

METHODS

Comment 5:

Methods p.5, listing numerous derived values that are not further presented or discussed or used to compare algorithms is unnecessary. Please delete all irrelevant measures that are not used for the comparison of the 3 methods.

Answer to Comment 5:

Assuming that perfusion parameters are addressed, we deleted the description of TTS, TDS, rCMRO2, and COV. Maps of OEF and CTH are presented in Figures 3 and 4.

Comment 6:

P3, line 89: “DCI was defined as cerebral infarction” Was there correspondence between hypoperfused region of brain and neurological abnormality? Please state explicitly.

Answer to Comment 6:

This point would be interesting, but needs to be addressed in a prospective study. The retrospective nature of this study limited the assessment of neurological deficits and DCI. Clinical records are imprecise for short-term symptoms. This may affect the internal validity and reproducibility of these findings. We therefore did not investigate the neurological symptoms in detail in our analysis.

Comment 7:

How was the core volume in the plain CT quantified?

Answer to Comment 7: 

We changed the standard to the infarct in the final CT. Please, see the answer to comment 3.

STATISTICS

Comment 8:

Statistics – the authors indicate that a Welsh test was used, but it is not clear if this is Welsh test for 2 groups or for 3 or more groups (Welsh ANOVA). Please clarify.

When analyzing data from 3 or more groups with unequal variances, the standard ANOVA test is not appropriate. Instead, robust alternatives like Welch's ANOVA or Brown-Forsythe's test should be used. These tests do not assume equal variances and are more reliable when dealing with heteroscedasticity (unequal variances).

The authors used Dunnett's post hoc test but this test assumes equal variances across the groups being compared. The Games-Howell post hoc test, like Welch’s analysis of variance, does not require the groups to have equal standard deviations.

Answer to Comment 8:

You are right. A Welch’s ANOVA was used for the analysis. We termed it wrong. We performed Games-Howell post hoc test and exchanged the values in the result section.

Comment 9:

Section 3.3 in Results should be placed in Methods/statistics.

 Answer to Comment 9:

Section 3.3 is integrated into the statistical analysis.

FIGURES AND LEGENDS

Comment 10:

All the legends are very poorly presented, essentially unintelligible or uninformative. In the legends, the letter that identifies each panel is sometimes at the beginning, other times in the middle of the description, the ROI that is colored is not clearly associated with the panel designated by the letter, and descriptions are very brief, incomplete and overall uninformative. This is all unnecessarily confusion and annoying. Please edit all figure legends extensively to improve clarity.

Answer to Comment 10:

 We apologize for the confusing descriptions. All legends are edited. The original colours of the software of Cercare (yellow/violet) and ISP (green/red) were used in the original version of the manuscript. We adapted all hypoperfused volumes to yellow areas and all core volumes to violet areas.

Figure 1 has been replaced to illustrate the intergroup comparison by the Welch ANOVA. Figure 2 is new and presents the treatment influence on the prediction of core and hypoperfused volumes.

Comment 11:

Fig. 2 figure legend is poorly written /confusing. “hypoperfused volumes (yellow/green) calculated by (C) Threshold; (D) Artificial Intelligence; (E) Intellispace Portal;” Here, yellow/green seems to belong to (D), not (C). Also, why do the authors show yellow ROIs only in (D) – why not show the ROI for the 3 methods for visual comparision? This is quite confusing.

Answer to Comment 11: Figure 2 is now Figure 3. Revised legend:

Figure 4: Case 1. Multimodal CT on day 9 after SAH (A, C-J) and follow-up plain CT on day 11 after SAH (B) are presented. The Artificial intelligence algorithm (D) calculated a huge hypoperfused area in the right MCA territory (yellow), with increases in the time to maximum (H) and a decrease in the oxygen extraction fraction (I). The remaining perfusion maps, regional cerebral blood flow (F), mean transit time (G), and capillary transit time heterogeneity (J) did not display noticeable changes. The Threshold (C) and Intellispace Portal (E) algorithms calculated no tissue at risk in the right MCA territory—no cerebral ischemia developed in the follow-up CT (B). CT = computer tomograph, MCA = middle cerebral artery, SAH = subarachnoid hemorrhage

Comment 12:

Fig 3 legend is a similarly confusing, unclear presentation referring to colored ROI in a B&W scan and not being clear about the time of the scans

Answer to Comment 12:

Figure 3 is now Figure 5. The description of Fig. 5 was revised as follows:

Figure 5: Case 2. Multimodal CT on day 13 after SAH (A, C-J) and follow-up plain CT one month after SAH (B) are shown. The algorithm of Artificial Intelligence (D) and Intellispace Portal (E) calculated a huge hypoperfused area in the left MCA territory (yellow), with increases in the mean transit time (G) and time to maximum (H). The oxygen extraction fraction (I) displayed an increase. Capillary transit time heterogeneity (J) showed only slight increases in the border zones, while relative cerebral flow did not display noticeable changes in the left MCA territory. No cerebral ischemia developed in the follow-up CT (B). CT = computer tomograph, MCA = middle cerebral artery, SAH = subarachnoid hemorrhage

Comment 13:

Fig. 4 is not referred to in the Results section and Fig 5 is referred to only in the Discussion. What is the point of these figures? They seem entirely superfluous.

 Answer to Comment 13:

Figures 4 and 5 are removed from the manuscript.

CASES

Comment 14:

Section 3.6 case 1 “ruptured aneurysms of the left PICA and the right MCA” It is very unusual, almost unheard of, to have two aneurysm rupture at the same time in the same patient. It is much more likely that one aneurysm ruptured and the other one was found incidentally. The presentation with right-sided hemiparesis suggests that the R MCA aneurysm may not be the one that bled, but later the patient was found to have severe narrowing of the R MCA. All of this is quite atypical, most unusual. I wonder if there’s an error in this presentation. The CT at presentation to show the pattern of hemorrhage may help explain things.

 Answer to Comment 14:

The PICA aneurysm was the most likely to have ruptured. The MCA aneurysm was an incidental finding, as you already assumed. Both aneurysms were coiled in the acute phase. The distribution of blood in the right basal cisterns around the proximal MCA branches may explain the following vasospasm of the right MCA. We amended the case description and added Figure 3 with the MRI and DSA at presentation.

Comment 15:

In Fig. 5, showing “time to maximum” is pointless. This was not a variable that was used to compare the 3 methods, and it conveys no useful information about the algorithms. There is no explanation of why it is shown.

Answer to Comment 15:

The commonly used thresholds for acute ischemic stroke are relative cerebral blood flow <30% for core volume and time to maximum >6s for hypoperfused tissue detection, which are thresholds specifically developed for acute ischemic stroke. Both perfusion parameters are essential to describe to define non-viable tissue and tissue at risk.

Figure 5 is deleted from the manuscript.

Reviewer 3 Report

Comments and Suggestions for Authors

Major Concerns

1. Lack of Diffusion-Weighted Imaging (DWI) as Gold Standard

The authors rely exclusively on non-contrast follow-up CT to define infarct core. While understandable in a retrospective study, this approach is suboptimal—particularly in DCI, where CT has limited sensitivity to early infarction and may miss cortical or small-volume ischemia. The absence of DWI significantly limits the precision of the ground truth against which CTP performance is measured.

Recommendation: This limitation must be explicitly acknowledged and discussed in depth. The authors should detail how infarct volume was segmented on CT, clarify interrater variability, and discuss the potential for false negatives, especially in non-comatose patients.

2. Extremely Divergent Sensitivity and Specificity Across Algorithms

The diagnostic performance across the three algorithms is strikingly inconsistent:

  • Cercare AI achieves 100% sensitivity but only 4.4% specificity.

  • ISP similarly shows 97.1% sensitivity but 0% specificity.

  • Threshold balances the two better but still with modest specificity (35.7%).

This raises substantial concern over the clinical applicability of these algorithms. The apparent overestimation of hypoperfused and infarct core regions by AI and ISP—driven perhaps by non-specific parameters such as TTP and rMTT—may lead to overtreatment and misclassification.

Recommendation: Authors should expand the discussion on why such discrepancies exist. Is it due to threshold selection, poor handling of artifacts (e.g., beam hardening), or lack of collateral flow modeling? A ROC analysis and spatial overlap metrics (e.g., Dice coefficient) should be considered to strengthen the comparison.

3. Assumption of Infarction as Inevitable Outcome in DCI

By equating hypoperfused regions with future infarction, the study assumes a deterministic pathophysiology. However, as the authors correctly note in cases like Case 1 and 2, many patients with low perfusion never progress to infarction, often due to interventions (e.g., induced hypertension or nimodipine). Therefore, using infarct volume as the definitive reference may undervalue the detection of “at-risk but salvageable” tissue.

Recommendation: Authors should discuss the risk of misclassifying reversible ischemia as false positives, and perhaps consider a three-category outcome: infarct, reversible ischemia, and no abnormality. Incorporating clinical outcome data (e.g., NIHSS or mRS) would be helpful if available.

Author Response

Response to Reviewer 3 Comments

Major Concerns

Comment 1:

  1. Lack of Diffusion-Weighted Imaging (DWI) as Gold Standard

The authors rely exclusively on non-contrast follow-up CT to define the infarct core. While understandable in a retrospective study, this approach is suboptimal—particularly in DCI, where CT has limited sensitivity to early infarction and may miss cortical or small-volume ischemia. The absence of DWI significantly limits the precision of the ground truth against which CTP performance is measured.

Recommendation: This limitation must be explicitly acknowledged and discussed in depth. The authors should detail how infarct volume was segmented on CT, clarify interrater variability, and discuss the potential for false negatives, especially in non-comatose patients.

Response to Comment 1:

Thank you very much for your effort and time to improve our manuscript with your ideas and meaningful comments.

We adopted the following sections:

Material and Methods:

“Standard

Infarct volume at follow-up CT was used as the standard measure. The plain CT scan at the time-point of CTP (baseline) was compared to previous CT scans to exclude prior posthemorrhagic or ischemic strokes from measurements. Then, ischemic brain lesions in baseline and follow-up CTs (following available CT scan) were delineated using a semi-automated technique (3D slicer, https://www.slicer.org/) by a neuroradiologist who was blinded to clinical data and perfusion maps. A region-of-interest thresholding with manual adjustments was used to segment hypodense lesions in high-resolution CT scans with a slice thickness of 1.5 mm or less.”

Limitations:

“Plain CT scans were exclusively segmented via 3D slicer to define final infarct volumes owing to clinical practice. MRI scan capacity has to be distributed among several competing departments. High-field MRI has additional limitations, the main ones being susceptibility to ferromagnetic material or contraindications due to the equipment needed by an intensive care unit patient during image acquisition. Therefore, the radiological monitoring was realized via multimodal CT for the question of DCI. Cortical and small-volume ischemia may be missed in DCI detection in this approach. This limits the precision of the ground truth against which the CTP performance is measured. Additionally, suboptimal scan quality can lead to interrater variability. 3D slicer is a user-friendly, open-source software that provides thresholding for segmentation, but needs visual control and adjustment.”

Comment 2:

  1. Extremely Divergent Sensitivity and Specificity Across Algorithms

The diagnostic performance across the three algorithms is strikingly inconsistent:

  • Cercare AI achieves 100% sensitivity but only 4.4% specificity.
  • ISP similarly shows 97.1% sensitivity but 0% specificity.
  • Threshold balances the two better but still with modest specificity (35.7%).

This raises substantial concern over the clinical applicability of these algorithms. The apparent overestimation of hypoperfused and infarct core regions by AI and ISP—driven perhaps by non-specific parameters such as TTP and rMTT—may lead to overtreatment and misclassification.

Recommendation: Authors should expand the discussion on why such discrepancies exist. Is it due to threshold selection, poor handling of artifacts (e.g., beam hardening), or lack of collateral flow modeling? A ROC analysis and spatial overlap metrics (e.g., Dice coefficient) should be considered to strengthen the comparison.

Response to Comment 2:

  1. We already mentioned the missing Dice coefficient in the limitations, which is caused by the limited data extraction of three different software packages. Lim et al. analysed 48 studies on automated CTP detection in acute ischemic stroke, of which only 4 performed a Dice coefficient analysis, mainly using one software package. Rava et al. was the only study found that compared three different software packages by analysing the overlap between the segmented areas and the ground truth.

Lim NE, Chia B, Bulsara MK, Parsons M, Hankey GJ, Bivard A. Automated CT Perfusion Detection of the Acute Infarct Core in Ischemic Stroke: A Systematic Review and Meta-Analysis. Cerebrovasc Dis. 2023;52(1):97-109. doi: 10.1159/000524916. Epub 2022 Jun 3. PMID: 35661075.

Rava RA, Snyder KV, Mokin M, Waqas M, Zhang X, Podgorsak AR, Allman AB, Senko J, Shiraz Bhurwani MM, Hoi Y, Davies JM, Levy EI, Siddiqui AH, Ionita CN. Assessment of computed tomography perfusion software in predicting spatial location and volume of infarct in acute ischemic stroke patients: a comparison of Sphere, Vitrea, and RAPID. J Neurointerv Surg. 2021 Feb;13(2):130-135. doi: 10.1136/neurintsurg-2020-015966. Epub 2020 May 26. PMID: 32457224.

  1. The CTP datasets were additionally divided into two groups for further analysis: one receiving “DCI-related treatment” and one “without DCI-related treatment”. Shown under comment 3.
  2. The discussion part was expanded as follows:

“Commonly used CT perfusion thresholds include rCBF <30% for core volume and Tmax >6s for hypoperfused tissue [16,17]. These thresholds were specifically designed for acute ischemic stroke. They slightly underestimate the ischemic core infarct volume, thereby favoring inclusion for treatment [18,19]. As highlighted by Chung et al. [20], the easy accessibility of CTP images overshadows the complexity of the underlying computational postprocessing workflow. It may lead to overconfidence in interpretation, thereby favoring diagnostic errors.

In our study, three algorithms with divergent thresholds were compared to identify the best-fitting algorithm for DCI. Previous studies evaluated arrival times [21] and different thresholds [22] for ISP in predicting final infarct size in ischemic stroke. A poor volumetric agreement was found for all methods in patients with complete re-canalization. Koopman et al. [23] compared ISP, RAPID, and syngo.via being superior in categorizing patients with small core infarcts. To our knowledge, there is no study published comparing Cercare Threshold or Cercare AI to other software packages.

The Cercare Threshold algorithm, applying the commonly used CT perfusion thresholds, derived the highest specificity for hypoperfused and core volumes compared to the final infarct volumes in follow-up examinations. It shows tissue highly endangered of hypoperfusion. The AI method calculated a larger hypoperfused volume than the Threshold method with Tmax >6s. One of the main reasons is that the AI model is based on TTP outlines (created by clinical experts), along with other maps, and not Tmax, as in the Threshold method. TTP lesions are larger than the Tmax lesions and thus correspond more to all affected tissue. At the same time, TTP is more susceptible to low cardiac ejection fraction, proximal stenosis, and poor bolus injections. In the treatment-based approach, core volumes of all algorithms corresponded more to the final infarct volumes, underlining the success of intra-arterial spasmolysis and induced hypertension in preventing infarct growth. The hypoperfused volumes did not become ischemic infarcts and were therefore overestimated. Rava et al. [24] described the same phenomenon for patients undergoing thrombectomy or not.

Correct prediction of ischemic core and penumbra in stroke was crucial for patient selection before randomized controlled trials investigated large core infarcts for mechanical thrombectomy [25]. With CTP being applied in different clinical scenarios, physicians must understand that using different software packages may produce different results that can impact their decisions. In DCI, there is no clear threshold indicating whether treatment is necessary or not [26]. Lolli et al. reviewed “early” (<72 h from aSAH) and “late” (>72 h from aSAH) CTP imaging, with aSAH being more informative when performed in a later time window. Lower tissue at risk thresholds (i.e., Tmax > 4s) were supposed in aSAH compared to acute ischemic stroke. Likewise, the importance of CTP on admission as an individual baseline reference for subsequent CTP studies was emphasized. Automated measurements from voxelwise analyses presented major limitations in aSAH due to the high incidence of DCI-unrelated parenchymal damage. All algorithms in our analysis included acute ICH in the calculated hypoperfused and volumes. Furthermore, intracranial devices are responsible for severe artefacts that render automated measurements.

 If no DCI-related treatment was performed, the core volumes were underestimated because tissue at risk of hypoperfusion progressed to ischemia. DCI is a progressive disease that can aggravate. Different from vessel occlusion in ischemic stroke, the absence of a clear perfusion deficit does not necessarily mean there is no tissue at risk. Sanelli et al. [27] found that aSAH patients with >50% angiographic narrowing of a given artery tend to have a perfusion deficit with reduced CBF. Contradictory results reported that patients with large or medium-sized arteries had vasospasm without having any perfusion deficit [28,29]. In the dynamic nature of vasospasm, it can regress in one territory following selective treatment on one CTP scan while at the same time affecting a new territory, which is difficult to quantify in a method that is based on a side-by-side comparison in a bilateral pathology. Beyond that, cortical spreading depolarization and seizures are accompanied by transitory hyperperfusion [12], which is not evaluated by the existing software packages. The evaluated three algorithms showed an incredibly high sensitivity of Cercare AI and ISP in detecting hypoperfused tissue, with an extremely poor specificity of <5%. These findings indicate limited evidence for clinical use of Cercare AI and ISP to predict tissue at risk of DCI. Cercare AI is superior in predicting core volumes independent of treatment. Cercare Threshold performed overall the best by applying the commonly used ischemic stroke thresholds. “

Comment 3:

  1. Assumption of Infarction as Inevitable Outcome in DCI

By equating hypoperfused regions with future infarction, the study assumes a deterministic pathophysiology. However, as the authors correctly note in cases like Case 1 and 2, many patients with low perfusion never progress to infarction, often due to interventions (e.g., induced hypertension or nimodipine). Therefore, using infarct volume as the definitive reference may undervalue the detection of “at-risk but salvageable” tissue.

Recommendation: Authors should discuss the risk of misclassifying reversible ischemia as false positives, and perhaps consider a three-category outcome: infarct, reversible ischemia, and no abnormality. Incorporating clinical outcome data (e.g., NIHSS or mRS) would be helpful if available.

Response to Comment 3:

  1. Thank you for the idea of a new categorization.

We chose four categories to calculate sensitivity, specificity, PPV, and NPV:

  • No perfusion deficit/no infarct development: no abnormality
  • No perfusion deficit/infarct development: underestimated progressive infarct
  • Perfusion deficit/no infarct development: reversible perfusion deficit
  • Perfusion deficit/infarct development: progressive infarct

Hypoperfused and core volumes were grouped according to volumes of 0 mL or greater. Final infarct volumes of the follow-up CTs were equally grouped into volumes of 0 mL or greater. For nine CTP datasets, there was no follow-up CT available, which had to be excluded from the analysis.

Tables 2, S1, and S2 present the calculated values in detail.

  1. Clinical outcomes are difficult to assess correctly in a retrospective study. Initial hemorrhagic stroke components, such as intracerebral hemorrhage, influence the clinical outcome in a substantial part. Moreover, clinical outcome is a patient-related outcome parameter which is not compatible with single CTP datasets. The resulting infarct volume can be measured retrospectively between two or more perfusion examinations. For that reason, final infarct volumes were chosen as standard and predictable outcomes.

  1. Based on ischemic stroke perfusion analyses, we additionally performed a therapy-dependent analysis of the datasets. After 51 multimodal CTs, the patients did not undergo DCI-specific therapy, as intra-arterial spasmolysis or induced hypertension. After 56 multimodal CTs, the patients underwent DCI-specific therapy. Intra-arterial spasmolysis and induced hypertension equivalent are rated in the Guidelines. Sixteen multimodal CTs were used as progress monitoring for patients who received intra-arterial spasmolysis shortly before but not after. These datasets were excluded from therapy-specific analysis.

Figure 2 and the following parts expanded the results:

“3.5 Treatment-related tissue prediction

All patients received standard DCI-treatment with Nimodipine orally (6 × 60 mg/day) or via gastric tube. Fifty-six CTP examinations were additionally followed by intra-arterial spasmolysis or induced hypertension. Sixteen examinations could not be grouped into “treatment” or “no treatment” due to prior intra-arterial spasmolysis being excluded for further investigations. Fifty-one CTP examinations remained without further DCI-related treatment. The distribution of the mean values for core and hypoperfused volumes compared to final ischemic infarct volumes is displayed in Figure 2. The four-category distribution is shown for DCI-related therapy in Table S1 and no therapy in Table S2.

DCI-related treatment

In the treatment-group (n=56), the Welch ANOVA calculated a significant deviation of the mean values for core volumes (p=0.009) and a highly significant deviation for hypoperfused volumes (p<0.001).

Core volumes calculated by Cercare Threshold (p=1.000), Cercare AI (p=0.260), and ISP (p=0.031) to predict non-viable tissue better corresponded to the final ischemic in-farct volume if a DCI-treatment was induced. The resulting infarct volumes were smaller compared to the group without treatment (Fig. 2), verifying treatment success. Therefore, the calculated hypoperfused volumes significantly deviated from the infarct volumes for all algorithms (Cercare Threshold: p=0.014, Cercare AI: p<0.001, ISP: p<0.001).

Cercare AI performed the best in predicting already non-viable tissue with a sensitivity of 73.7% and a specificity of 51.4% (Table S1). Considering the prediction of infarct development, the sensitivity increased for Cercare Threshold in the prediction of tissue at risk, with 85.7%. The specificity of Cercare AI and ISP remained low despite a 100% sensitivity.

No DCI-related treatment

In the group without treatment (n=51), means for both core and hypoperfused volumes (p<0.001) deviated highly significantly from the final ischemic infarct. Without treatment, the final infarct exceeded the predicted core volumes, underlining the need for DCI-specific treatment (Fig. 2). The calculated hypoperfused volumes now better corresponded to the final infarct size (Fig. 2). There was no significant deviation of the infarct volume and the calculated core (Cercare Threshold: p=0.057, Cercare AI: p=0.156, ISP: 0.280) or hypoperfused volumes (Cercare Threshold: p=0.424, Cercare AI: p=0.937, ISP: p=0.569) of the algorithms due to the small dataset. Only the volumes of Cercare Threshold and ISP had significant intergroup variance (p<0.001) for both per-fusion volumes.

Without DCI-related treatment, the sensitivity of core prediction rose for Cercare AI (87.5%) and ISP (93.8%), with a specificity of 62.9% for Cercare (Tab. S2) AI. ISP presented an alarmingly low specificity in the prediction of non-viable tissue (8.6%) and tissue at risk (0.0%). Cercare Threshold presented overall the best specificity with 77.1% for non-viable tissue and 51.4% for tissue at risk of hypoperfusion.”

Round 2

Reviewer 2 Report

Comments and Suggestions for Authors

This version is much improved. I have no further critiques.

Author Response

Comment: This version is much improved. I have no further critiques.

Answer to Reviewers Comment: Thank you very much for your helpful comments!

Reviewer 3 Report

Comments and Suggestions for Authors

Thank you for the thoughtful and well-structured revision. The manuscript presents a clinically relevant study evaluating CT perfusion postprocessing algorithms in the setting of delayed cerebral ischemia (DCI) after subarachnoid hemorrhage. The authors have clearly improved the manuscript by addressing several key concerns:

  • The use of 3D Slicer for semi-automated infarct segmentation, including the description of thresholding and manual adjustment protocols, enhances methodological transparency. The rationale for not using DWI due to ICU constraints and MRI contraindications is well justified and appropriately discussed.

  • The introduction of a four-category classification framework (including reversible perfusion deficits) is a significant improvement that adds clinical nuance. The treatment-stratified subgroup analysis further strengthens the interpretation of perfusion-infarct mismatch in DCI.

That said, one important methodological limitation remains. The lack of ROC curve analysis or spatial validation metrics (e.g., Dice coefficient) weakens the quantitative comparison between algorithms. While we acknowledge the technical constraints associated with proprietary software, we strongly encourage the authors to either:

  1. Perform a subset ROC or spatial overlap analysis on available data, or

  2. Explicitly expand the discussion of how the absence of these metrics may affect interpretation, generalizability, and reproducibility of the findings.

Minor clarifications in the methods section (e.g., interrater reliability) and a brief contextualization of your findings in relation to acute stroke CTP validation studies would further strengthen the manuscript.

Overall, this is a valuable contribution that helps refine the clinical interpretation of perfusion imaging in DCI. With minor revisions, the work will be suitable for publication.

Author Response

Comment 1:

Thank you for the thoughtful and well-structured revision. The manuscript presents a clinically relevant study evaluating CT perfusion postprocessing algorithms in the setting of delayed cerebral ischemia (DCI) after subarachnoid hemorrhage. The authors have clearly improved the manuscript by addressing several key concerns:

  • The use of 3D Slicer for semi-automated infarct segmentation, including the description of thresholding and manual adjustment protocols, enhances methodological transparency. The rationale for not using DWI due to ICU constraints and MRI contraindications is well justified and appropriately discussed.
  • The introduction of a four-category classification framework (including reversible perfusion deficits) is a significant improvement that adds clinical nuance. The treatment-stratified subgroup analysis further strengthens the interpretation of perfusion-infarct mismatch in DCI.

Answer to Comment 1:

Thank you for esteeming the revised manuscript.

Comment 2:

That said, one important methodological limitation remains. The lack of ROC curve analysis or spatial validation metrics (e.g., Dice coefficient) weakens the quantitative comparison between algorithms. While we acknowledge the technical constraints associated with proprietary software, we strongly encourage the authors to either:

  1. Perform a subset ROC or spatial overlap analysis on available data, or
  2. Explicitly expand the discussion of how the absence of these metrics may affect interpretation, generalizability, and reproducibility of the findings.

Answer to Comment 2:

We expanded the discussion as follows:

This study lacks the evaluation of spatial resolution, e.g., the Dice similarity coefficient. Comparison of spatial resolution requires the generation of 3D contours for co-registration of infarct volumes and calculated hypoperfused and core volumes. Same volume measurements can be acquired in different brain regions without any spatial overlap. Our analysis cannot precisely differentiate between artefact detection (e.g. skull base), correct infarct prediction, and new infarct development outside the perfusion deficit.

We did not perform a Receiver Operating Characteristics (ROC) analysis because of a missing graduated thresholding factor (e.g. Tmax). Two software tools, each with its own generated perfusion maps, were evaluated. Each of these used divergent AIF, VOF, and artefact corrections. The three algorithms to calculate hypoperfused and core volumes had divergent underlying parameters (ISP: rMTT and CBV, Cercare Threshold and AI: rCBF and Tmax). These preconditions are inappropriate for performing a ROC analysis. Moreover, binary classifiers are statistical and computational models that divide a dataset into two groups, positives and negatives. ROC’s primary limitation is that the ROC curve can be skewed when the classes in a study are imbalanced. Two of the three algorithms clearly overestimated the hypoperfused volumes, resulting in low specificity.  The visual interpretability of ROC curves in the context of imbalanced datasets can be deceptive regarding conclusions about the reliability of classification performance, due to an intuitive but incorrect interpretation of specificity.

Comment 3:

Minor clarifications in the methods section (e.g., interrater reliability) and a brief contextualization of your findings in relation to acute stroke CTP validation studies would further strengthen the manuscript.

Answer to Comment 3:

We expanded the limitations as follows:

The automated selection of AIF and VOF for Cercare algorithms increases the interrater reliability. Whereas these basic functions need to be manually selected in the Intellispace Portal software. Determining the terminal segment of the ICA (AIF) and superior sagittal sinus (VOF) as a target was the applied rule, which can vary between raters in clinical practice. In this study, only one neuroradiologist generated the analysed perfusion maps applying the same rules overall.

The discussion was amended as follows:

As highlighted by Demeestere et al. [21], the infarcted tissue will not be identified as nonviable tissue on CTP if CBF exceeds the threshold for ischemic core detection. The inspection of the plain CT for subacute or established infarct development is thus mandatory, especially in patients with SAH. CTP has a relatively limited spatial resolution and, consequently, a limited sensitivity in detecting lacunar or small subcortical infarcts. This resulted in a sensitivity of only 62% compared with diffusion-weighted magnetic resonance imaging in another study [22]. Sensitivity for infratentorial lesions is, however, low.

Overall, this is a valuable contribution that helps refine the clinical interpretation of perfusion imaging in DCI. With minor revisions, the work will be suitable for publication.
